# Rapid diagnosis and reduced workload for urinary tract infection using flowcytometry combined with direct antibiotic susceptibility testing

Hanne Margrethe Gilboe◯*, Olaug Marie Reiakvam[¤a], Linda Aasen[¤b], Trygve Tjade, Johan Bjerner, Trond Egil Ranheim, Peter Gaustad◯[¤c]

Department of Microbiology, Fürst Medical Laboratory, Oslo, Norway

¤a Current address: Department of Microbiology, Oslo University Hospital, Oslo, Norway
¤b Current address: Department of Medical Genetics, Oslo University Hospital, Oslo, Norway
¤c Current address: University of Oslo, Oslo, Norway
* hmgilboe@furst.no

## Abstract

### Background

We evaluated if flowcytometry, using Sysmex UF-5000, could improve diagnosis of urinary tract infections by rapid identification of culture negative and contaminated samples prior to culture plating, thus reducing culture plating workload and response time. We also evaluated if it is possible to reduce the response time for antibiotic susceptibility profiles using the bacteria information flag on Sysmex UF-5000 to differentiate between Gram positive and negative bacteria, followed by direct Antibiotic Susceptibility Testing (dAST) on the positive urine samples.

### Methods

One thousand urine samples were analyzed for bacteria, white blood cells and squamous cells by flowcytometry before culture plating. Results from flowcytometric analysis at different cut-off values were compared to results of culture plating. We evaluated dAST on 100 urine samples that were analyzed as positive by flowcytometry, containing either Gram positive or Gram negative bacteria.

### Results

Using a cut-off value with bacterial count $\geq$100.000/mL and WBCs $\geq$10/µL, flowcytometry predicted 42,1% of samples with non-significant growth. We found that most contaminated samples contain few squamous cells. For 52/56 positive samples containing Gram negative bacteria dAST was identical to routine testing. Overall, there was concordance in 555/560 tested antibiotic combinations.

### Conclusion

Flowcytometry offers advantages for diagnosis of urinary tract infections. Screening for negative urine samples on the day of arrival reduces culture plating and workload, and results in

**Data Availability Statement:** All relevant data are within the manuscript and its Supporting Information files.

**Funding:** The author(s) received no specific funding for this work.

**Competing interests:** The authors have declared that no competing interests exist.

shorter response time for the negative samples. The bacteria information flag predicts positive samples containing Gram negative bacteria for dAST with high accuracy, thus Antibiotic Susceptibility Profile can be reported the day after arrival. For the positive samples containing Gram negative bacteria the concordance was very good between dAST and Antibiotic Susceptibility Testing in routine. For positive samples containing Gram positive bacteria the results were not convincing. We did not find any correlation between epithelial cells and contamination.

## Introduction

Urinary tract infections (UTI) are one of the most common bacterial infections with 50–60% lifetime incidence in adult women [1,2]. Prompt diagnosis and correct antibiotic treatment is desirable to avoid severe complications [3] and reduce antibiotic resistance.

Diagnosis of UTI is based on clinical symptoms as well as finding pyuria and bacteriuria. Urine culture is currently the gold standard for the diagnosis of UTI [4,5] but is labor intensive with a turnaround time of 24–48 hours. Urine culture is often a significant workload and time-consuming procedure in clinical microbiology laboratories.

Sysmex UF-5000 is a third generation fully automated flowcytometer. It analyzes and classifies cells by analyzing forward scatter light (FSC), side scatter light (SSC), side fluorescent light (SFL) and depolarized side scattered light (DSS) [6–8]. Of the 17 parameters available for urine analysis on Sysmex UF-5000 total bacterial count (TBC), white blood cell (WBC) count and number of epithelial cells are most useful in UTI diagnostics. The bacteria information flag (bact. flag) provides information about the type of bacteria detected in the urine samples (Gram negative, Gram positive, combinations of the two or unspecified).

We evaluated if flowcytometry could improve UTI diagnosis by rapid identification of culture positive, culture negative and contaminated samples prior to culture plating based on the 4 parameters: 1. WBC count, 2. bacterial count, 3. Gram positive or Gram negative bacteria, 4. squamous cells count, and if this could reduce workload associated with culture plating and time to results for culture negative and contaminated samples.

We also evaluated if it is possible to reduce the response time for the antibiotic susceptibility profiles for culture positive samples using the bact. flag on Sysmex UF-5000 to differentiate between Gram positive and negative bacteria, followed by direct Antibiotic Susceptibility Testing (dAST) on the urine samples that are analyzed as positive and containing either Gram positive or Gram negative bacteria. This could reduce response time for Antibiotic Susceptibility Profiles (ASP) by 24 hours.

## Materials and methods

### Study design and samples

One thousand urine samples were selected on five separate days over a period of three weeks between 2017-09-14 and 2017-10-03. Transportation times to the laboratory were monitored, and 608 samples (61.1%) were received at the laboratory same day as they were voided, 303 samples (30.5%) the day after they were voided, 23 samples (2.3%) the second, 47 samples (4.7%) the third day and 4 samples (0.4%) the fourth day after the day they were voided. Transportation time was not recorded for 10 samples (1.0%). All samples that were selected arrived at the lab in recommended 4 mL sterile containers (Becton Dickinson Vacutainer Plus C&S

Boric Acid, REF 364959) containing boric acid. All the randomly selected urines contained adequate volume for both cultivation and flowcytometric analysis. Fürst Medical Laboratory recommends culture plating and examination of urine samples in boric acid within 48 hours after voiding, in concordance with European Urinalysis Guidelines [16]. However, as transportation time from rural parts of Norway can be delayed, especially on weekends and holidays, samples are considered for culture plating and examination if there is information about clinical infection, or/and antibiotics has been initiated, as these samples are difficult to reproduce. If the urine sample is culture plated and examined, the clinician will receive a comment that the result is uncertain as the sample was not culture plated and examined within the recommended time frame.

Samples were from outpatients and requested by either General Practitioners (984 samples) or from nursing homes (11 samples). 819 samples (82.3%) were from women and 176 samples (17.7%) from men. Furthermore, 21 samples (2.1%) where from subjects aged 0–4 years, 46 samples (4.6%) from subjects aged 5–14 years, 403 samples (40.5%) from subjects aged 15–44 years, and finally 525 samples (46.2%) from subjects aged 45 years and older. All urine samples were analyzed on Sysmex UF-5000 before routine culture plating. Results from the flowcytometric analysis were compared to the results of routine culture plating.

For the first part of the study, different cut-off values for bacteria and WBCs were evaluated, and the number of squamous cells was compared to culture results. For the second part of the study, randomly selected urine samples that were analyzed by flowcytometry to be positive and containing either Gram positive or Gram negative bacteria were tested by dAST. The selected cut-off values for bacteria and WBCs were bacterial count $\geq$10/mL and WBCs $\geq$ 10/ µL, based on previous studies and well-known guidelines regarding urine culturing [9]. The samples were analyzed using the bact.flag and categorized as Gram positive, Gram negative and mixed Gram positive/negatives. Selected positive urine samples containing either Gram positive or Gram negative bacteria were spread directly on agar plates containing antibiotic discs, incubated overnight, and interpreted manually the following day using guidelines established by the Norwegian Working Group on Antibiotics (AFA) [10]. The antibiotic resistance panel for the Gram negative bacteria consisted of Mecillinam, Ampicillin, Ciprofloxacin, Trimethoprim/sulfamethoxazole, Trimethoprim, Nitrofurantoin, Ceftazidime, Meropenem, Cefoxitin, Cefotaxime. The Gram positive resistance panel consisted of Penicillin, Trimethoprim/sulfamethoxazole, Trimethoprim, Nitrofurantoin, Amoxicillin. We compared the results of dAST with identification and routine Antibiotic Susceptibility Testing (AST), using automated bacterial identification and susceptibility testing (Microscan, Siemens) or Mass Spectrometry (MALDI-TOF (Matrix-assisted desorption ionization-time of flight mass spectrometry)) followed by manual AST. Discrepancies in the interpretation of susceptible (S), intermediate (I) and resistant (R) between the two methods were classified as minor error (mE), major error (ME) and very major error (VME). The mE included strains that were interpreted as S or I with one method and respectively I or R with the other method. The ME includes strains that were interpreted as R with direct AST and S with the routine AST, and the VME included strains that were interpreted as S with dAST and R with the routine AST.

### Flowcytometric analysis using Sysmex UF-5000

The Sysmex UF-5000 was used as recommended by the company (Sysmex Corporation). It uses a flowcytometry-based system, with forward scatter light (FSC), side scatter light (SSC) and side fluorescent light (SFL). In addition to these components, the Sysmex UF-5000 uses depolarized side scattered light (DSS) to differentiate between red blood cells and crystals. The machine is fully automated, has a modular concept for urinalysis workflow and can be used

with UD-10, a newly developed urine image viewer. The samples were analyzed on automatic mode with the recommended long rinse cycle switched on to avoid carryover between samples [11]. The bact. info flag provides information about the type of bacteria detected in the sample (Gram negative, Gram positive, combinations of the two or unspecified) based on different light signals of FSC, SFL and SHH for Gram positive and Gram negative bacteria, due to different dye intake by the cell wall structures.

## Urine culture

After the initial flowcytometric analysis all urine samples were routinely plated on chromatic biplates (Becker Dickinson CHROMagar Orientation Medium/Columbia CNA Agar) by an automated microbiology plater instrument (WASP Walk Away Specimen Processor, Copan Inc.) The culture plates were incubated at ambient atmosphere at $35^0$ C for 18 to 24 hours prior to manual interpretation. Samples were interpreted according to national guidelines [12] as illustrated in Table 1. Culture plated samples with significant growth (indicative of UTI), contained a significant number of one or (more rarely) two significant uropathogens. Samples with non-significant growth (not indicative of UTI) were either culture negative or mixed/contaminated and did not meet the criteria used for significant urine growth (Table 1). Mixed cultures contain 3 or more types of bacteria, either Gram positive, Gram negative or both. Further identification and AST for suspected E. coli was performed on Microscan. Other bacteria were further analyzed with MALDI-TOF for identification, followed by manual AST according to EUCAST guidelines [13]. Samples with non-significant findings were not subjected to further testing.

## Data analysis

Descriptive statistics were used to define the samples and compare the results from flowcytometry to cultivation. The sensitivity, specificity, and predictive values for different cut-off values for bacteria and leucocytes were calculated. Collected anonymized data were recorded on Microsoft Excel spreadsheets. Statistical analysis was performed using the Microsoft Excel (Windows, Microsoft Office Professional 2000).

**Table 1. Significant values for different uropathogens in different patient groups.** (Adapted from Norwegian national guidelines for diagnosing bacterial urinary tract infections) [4].

| Patient group | | Pure culture | | | Culture with 2 microbes |
|---|---|---|---|---|---|
| SymptomaticPatients(MSU and IC)* | Women | Primary[1] pathogens (CFU/ml) | Secondary[2] pathogens (CFU/ml) | Doubtful pathogens[3] (CFU/ml) | Primary and secondary pathogens (CFU/ml) |
| | | $\geq 10^3$ | $\geq 10^4$ | $\geq 10^5$ | $\geq 10^4$ *Staphylococcus saprophyticus* $\geq 10^3$ |
| | Men | $\geq 10^3$ | $\geq 10^4$ | $\geq 10^5$ | $\geq 10^4$ |
| | Children | $\geq 10^3$ | $\geq 10^4$ | $\geq 10^4$ | $\geq 10^4$ |

*Mid-stream urine and urine collected by intermittent catheterization.

[1] Must always be identified and subjected for antibiotic susceptibility testing (AST): *Escherichia. coli*, *Staphylococcus saprophyticus*, *Salmonella* spp.

[2] Must always be identified and subjected to AST: *Enterobacter* spp., *Enterococcus* spp., *Streptococcus gallolyticus*, *Klebsiella* spp., *Proteus mirabilis*, *Pseudomonas aeruginosa*. *Citrobacter* spp., *Morganella.morganii*, *Proteus vulgaris*, *Serratia* spp., *Staphylococcus aureus/argenteus*. *Corynebacterium urealyticum*, *Streptococcus pneumonia*.

[3] ID and AST carried out if clinically indicated: Streptococci gr.A,B,C,G, yeast, coagulase negative staphylococci (other than *Staphylococcus saprophyticus*), *Acinetobacter* spp., *Pseudomonas* spp., *Stenotrophomonas.maltophilia*, *Aerococcus urinae/sangunicola*. *Streptococcus urinalis*.

The following bacteria are considered apatogens in urine and should only in rare cases be identified and subjected to AST: α hemolytic Streptococcus, *Gardnerella. vaginalis*, Lactobacilli, *Bifidobacterium* spp., diphtheroid rods etc.

### Ethical approval

The study was reviewed approved by the Regional Committee for Medical and Health Research Ethics. As the study was outside the remit of the Act on Medical and Health Research 2008, formal ethical approval was not required. (IRB ref: IRB00001871). The need to obtain consent for the study was waived by the committee. All the urine samples were analyzed for possible UTI as ordered by the clinicians. Patient data were anonymized before flowcytometric and data analysis.

## Results

### Cultivation

Of the 1000 culture plated samples, 223 (22,3%) were culture positive, indicative of UTI and reported after identification and ASP (Table 2). Gram negative bacteria were the most frequent bacteria isolated from the samples with significant growth. *Escherichia coli (E.coli)* was the most frequent bacteria, accounting for 74,4% of all significant findings. In 777 (77,7%) samples there was non-significant growth, not indicative of UTI and showed either mixed growth (indicative of contamination (628 (62,8%)) or were culture negative (149 (14,9%)).

### Flowcytometric analysis of culture plated samples

As shown in Fig 1, bacteria was detected in almost all 1000 urine samples analyzed by flow cytometry. Only 0,8% contained bacteria < 1000/mL. The majority (72,7%) contained bacteria ≥ 100.000/mL, and 90,9% of the samples contained bacteria ≥10.000/mL. For the culture plated samples with significant growth 95,5% contained bacteria ≥ 100.000/mL and 99,1% contained bacteria >10.000/mL. Only 0,9% contained bacteria <10.000/mL. For the samples with significant growth 93,9% of the samples contained WBC ≥10/μL, 85,8% ≥ 20 WBC/μL, 75,3% ≥30 WBC/μL, 60,7% ≥ 40 WBC/μL. For the samples with non-significant growth 66,2% of the samples contained bacteria ≥100.00/mL, 89,2% contained bacteria ≥ 10.000/mL (10,8% <10.000), 99% contained bacteria ≥ 1000/mL. Only 1% contained bacteria < 1000/mL (Fig 2). 16% of the samples contain WBC <10/μL., 36,2% < 20 WBC/μL, 58,6% < 30 WBC/μL, and 82,2% < 40 WBC/μL. 17.1% contained WBC ≥ 40/μL.

### Evaluation of flowcytometric cut-off values compared to cultivation results

Different cut-off values for WBCs and bacteria on Sysmex UF-5000 were investigated and compared with results from culture plating to evaluate the number of significant

**Table 2. Identified bacteria in samples in with significant growth.**

| Bacterial species | No./% |
|---|---|
| *Escherichia coli* | 165/74.4 |
| *Enterococcus fecalis* | 21/9.4 |
| *Klebsiella* sp. | 16/7.2 |
| *Staphylococcus saprophyticus* | 9/4.0 |
| *Enterobacter* sp. | 6/2.7 |
| *Citrobacter* sp. | 2/0.9 |
| *Proteus mirabilis* | 2/0.9 |
| *Pseudomonas aeruginosa* | 1/0.4 |
| Group B streptococci | 1/0.4 |
| Total | 223/100 |

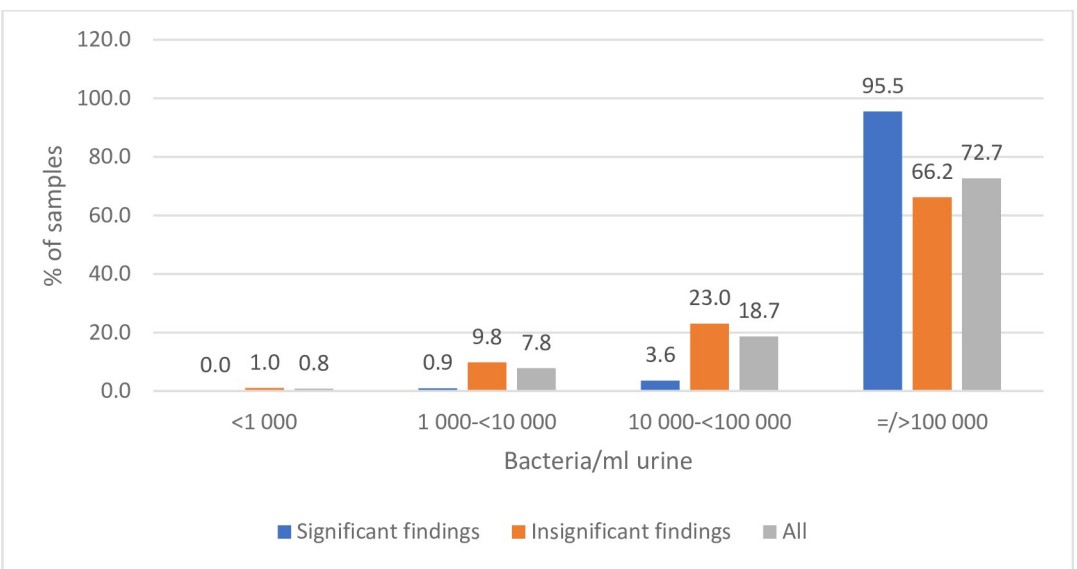

**Fig 1. Flowcytometric findings (bact./mL) in culture plated with significant growth (significant findings) and non-significant growth (non-significant findings).**

culture positive samples missed at each value (false negatives), and how many samples with insignificant growth could be detected (thus reducing culture plating), as shown in Table 3. Cut-off values from 10–100.000 bacteria/mL and 10–40 WBCs/uL were evaluated.

**False negative samples. Table 4 shows details of false negative samples using a cut-off value with** bacteria $\geq$100.000/mL and WBCs. $\geq$10/μL.

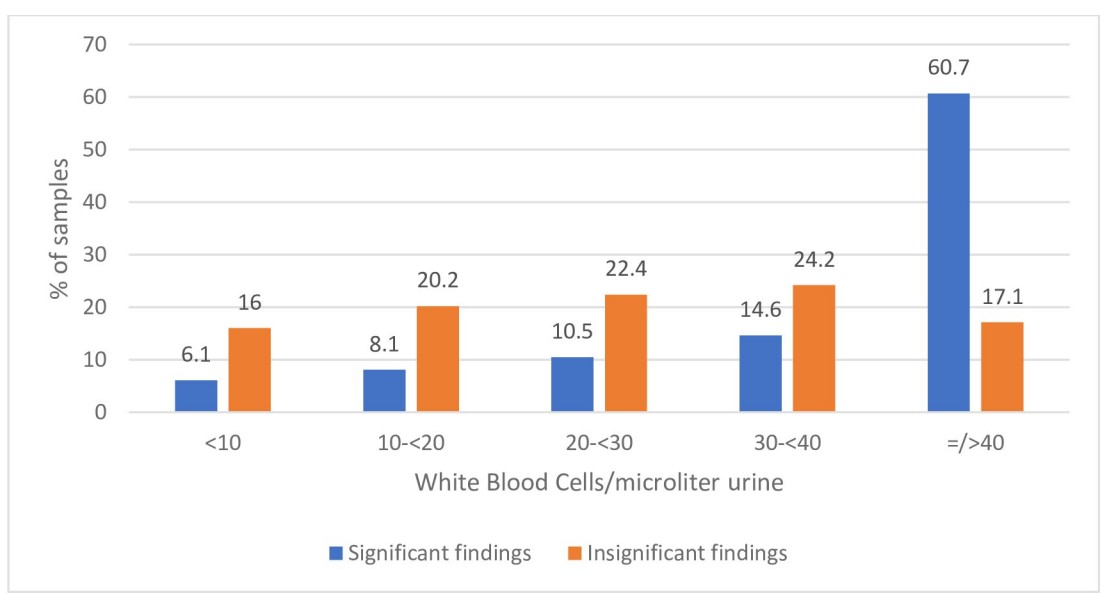

**Fig 2. Flowcytometric findings (WBCs/μl) in culture plated samples with significant growth (significant findings) and non-significant growth (non-significant findings).**

**Table 3. Evaluation of different cut-off values for WBCs and bacteria on flowcytometry.**

| Cut-off | Lost Samples with significant Growth. (false negative) | Samples with insignificant growth (reduced culture plating) | True «positives» | True «negatives» | False «positives» | Sens. | Spes. | PPV | NPV |
|---|---|---|---|---|---|---|---|---|---|
| | no./% | % | n | n | n | % | % | % | % |
| ≥10.000 bacteria (bact)/mL. | 2[1]/0,2 | 8,6 | 221 | 84 | 693 | 99,0 | 10,8 | 24,2 | 97,7 |
| ≥ 100.000 bact/mL. | 10[2]/1 | 27,3% | 213 | 263 | 514 | 95,5 | 33,8 | 29,3 | 96,3 |
| ≥10.000 bact./mL ≥ 10 white blood cells (WBCs)/µL | 19[3]/1,9 | 33,0% | 204 | 311 | 466 | 91,5 | 40,0 | 30,4 | 94,2 |
| ≥ 100.000 bact/mL. ≥ 10 WBCs./µL | 27[4]/2,7 | 42,1% | 196 | 394 | 383 | 87,8 | 50,7 | 33,8 | 93,5 |
| ≥ 10.000 bact./mL. ≥ 20 (WBCs)/µL | 25/2,5% | 41,1% | 198 | 386 | 391 | 88,8 | 49,7 | 33,6 | 93,9 |
| ≥ 100.000 bact/mL. ≥ 20 WBCs/µL | 30/3,0% | 47,4% | 193 | 391 | 333 | 86,5 | 54 | 36,7 | 92,9 |
| ≥ 10.000 bact/mL. ≥ 30 WBCs/µL | 32/3,2% | 45,4% | 191 | 422 | 355 | 85,6 | 54,3 | 35 | 93 |
| ≥ 100.000 bact/mL. ≥30 WBCs/µL | 38/3,8% | 50,9% | 186 | 472 | 305 | 83,4 | 60,7 | 37,9 | 92,7 |
| ≥ 10.000 bact/mL. ≥40 WBCs/µL | 44/4,4% | 50,1% | 179 | 457 | 320 | 80,3 | 58,8 | 35,9 | 91,2 |
| ≥100.000 bact/mL. ≥ 40 WBCs/µL | 49/4,9% | 54,6% | 174 | 497 | 280 | 78 | 64 | 38,3 | 91 |

[1] *E. coli* 1 *Enterocuccus* spp. 1.

[2] *E. coli* 7. *E. fecalis* 3.

[3] *E. coli* 13. *K. pneumoniae* 1. *E. fecalis* 5.

[4] *E. coli* 19 *K. pnemoniae* 1. *E. fecalis* 7.

PPV: Positive predictive value. NPV: Negative predictive value.

## Squamous cells -contamination

We investigated if detection of squamous cells by Sysmex UF-5000 could be used as a parameter for contaminated samples. As the Fig 3 illustrates, most of the contaminated samples contains very few squamous cells.

## Identification and dAST on the positive urine samples selected by flowcytometry

We evaluated if it is possible do dAST and identification directly on the urine samples that were analyzed as positive by flowcytometry, containing Gram positive or Gram negative bacteria.

**Positive samples by flowcytometry containing Gram negative bacteria.** As shown in Table 5, 18 of the selected 81 positive samples containing Gram negative bacteria were

**Table 4. Prevalence of bacteria in false negatives samples with cut-off value bacteria ≥100.000/mL and WBCs ≥10/µL.**

| | No. bacteria | % of the total identified bacteria | % of the total false negatives |
|---|---|---|---|
| *E. coli* | 19 | 11,5 | 70,4 |
| *E. fecalis* | 7 | 33,3 | 25,9 |
| *K. pneumoniae* | 1 | 6,3 | 3,7 |
| | 27 | | 100 |

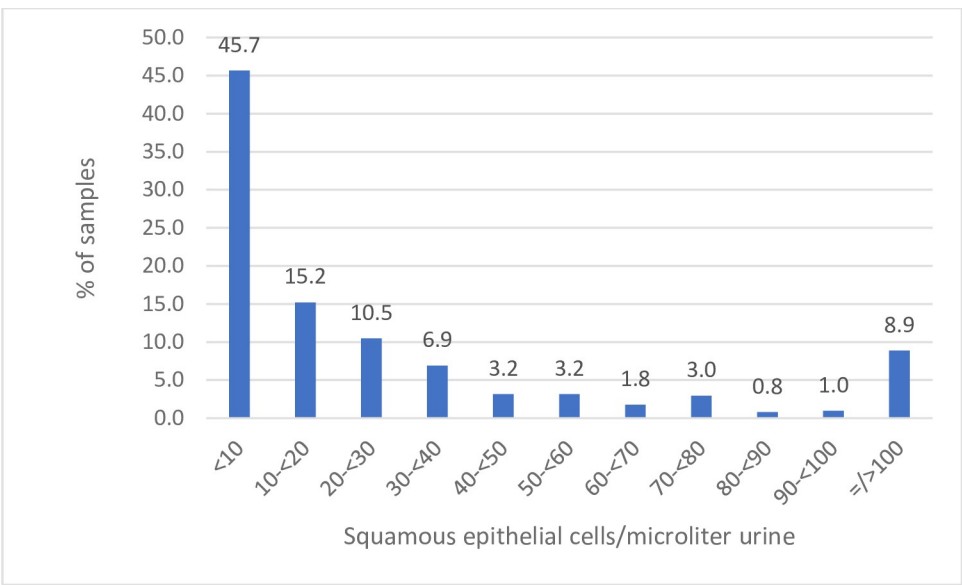

**Fig 3. Number of squamous cells in contaminated samples.**

excluded due to mixed growth when culture plated, and one due to growth of two species of bacteria resulting in 62 samples to evaluate AST. Due to technical problems two AST's were not successful and excluded, and 4 were excluded due to mixed growth/contamination for some of the antibiotic discs. In the remaining 56 samples, 52 samples showed full concordance between dAST and routine AST (Table 6). In the four samples without full concordance there was discrepancy for one antibiotic in 3 samples, and for two antibiotics in one sample. All the samples with discrepancies in antibiotics were identified as *E.coli*.

Of the 5 identified discrepancies we identified two mE, two ME one VME (as shown in Table 7). In one sample Trimethoprim (TMP) and Trimethoprim/sulfamethoxazole (SXT) were resistant on dAST and intermediate on routine AST. In two samples dAST were resistant for either Ampicillin (AM) or TMP but sensitive in routine AST. In one sample dAST was Ampicillin sensitive, but routine AST was Ampicillin resistant. This VME was, however, only detected for one of 560 antibiotics.

**Positive samples by flowcytometry containing Gram positive bacteria.** Of the 25 randomly selected samples that were positive on flowcytometry containing Gram positive bacteria, 14 samples were contaminated on culture plating and thus excluded. One sample was identified as Gram negative bacteria (*E.coli*), one sample as yeast, and three had no growth on the culture plates, leaving six samples available for interpreting AST, and comparing with dAST.

**Table 5. Culture results for 81 positive samples by flowcytometry containing Gram negative bacteria.**

| Sysmex UF-5000 Results | Culture Results | Comments | No. Samples |
|---|---|---|---|
| Gram negative Bacteria | Gram negative bacteria | *E. coli* >100 000 CFU/ml x 54, *K. pneumonia* >100 000 cfu/ml x 7, *P. aerguinosa* >100 000 CFU/ml x 1. | 62 |
| Gram negative Bacteria | Contaminated sample | | 18 |
| Gram negative Bacteria | 1 Gram negative + 1 Gram positive | Gram positive bacteria *E. faecalis* >100 000 CFU/ml. Gram negative bacteria *E. aspuriae* >100 000 CFU/ml. | 1 |
| Total Gram negative Bacteria | | | 81 |

**Table 6. Evaluation of 56 samples available for dAST.**

|  | No. | % |
|---|---|---|
| Available for AST | 56 | 100 |
| Discrepancy 1 AB | 3 | 3,7 |
| Discrepancy 2 AB | 1 | 1,2 |
| **AST Concordance** | **52 samples** | **84** |

Of the six samples one was discarded due to technical difficulties, one was identified as *Streptococcus hemolyticus* and discarded (unlikely cause of UTI and routine AST not indicated). Of the four remaining samples three were identified as *Staphylococcus saprophyticus*, one as *Staphylococcus. epidermidis*. For these samples there was full concordance between the dAST and routine AST.

## Discussion

### Flowcytometric results

Bacteria were present in almost all samples analyzed by flowcytometry, even if there was no growth on cultivation (Fig 1). Flowcytometry counts non-viable as well as viable bacteria. In addition, some of the viable bacteria does not grow in regular cultivation condition. The high number of bacteria in non-significant samples (66,2% contain $\geq$100.000 bacterial/mL), reflects the large number of contaminated samples, often containing many different bacteria. The culture negative samples are reflected in the 10,8% of the non-significant samples containing <10.000/mL. bacteria.

Our study found that WBCs are present in a higher number in the culture positive samples, but also present in samples with non-significant growth on cultivation. In a recent study by Cho et al. Sysmex UF-5000 was evaluated to have excellent performance for WBC and better performance than any other automated platform for bacteria [7].

Sterile pyuria is not an uncommon finding. In a paper by Peter Glen [14] 9% percent of patients presenting to their GP with lower urinary symptoms are found to have sterile pyuria. It can be classified as infectious or non-infectious and can be caused by sexually transmitted diseases [15], parasitic and atypical infections, previous surgical procedures or pelvic

**Table 7. Discrepancy of antibiotic susceptibility comparing routine AST to dAST.**

| Antimicrobial agent | No. of samples with dAST in concordance with routine AST. | Concordance % | Minor Error (mE) | Major Error (ME) | Very Major Error (VME) |
|---|---|---|---|---|---|
| Ampicillin | 54/56 | 96% |  | 1 sample R dAST, S routine AST. | 1 sample S dAST, R routine AST. |
| Mecillinam | 56/56 | 100% |  |  |  |
| Ciprofloxacin | 56/56 | 100% |  |  |  |
| Trimethoprim | 54/56 | 96% | 1 sample S dAST, I routine AST. | 1 sample R dAST, S routine AST. |  |
| Trimethoprim/ sulfamethoxazole | 55 | 98% | 1 sample S TMP dAST, I routine AST. |  |  |
| Nitrofurantoin | 56 | 100% |  |  |  |
| Ceftazidime | 56 | 100% |  |  |  |
| Meropenem | 56 | 100% |  |  |  |
| Cefoxitin | 56 | 100% |  |  |  |
| Cefotaxime | 56 | 100% |  |  |  |

radiotherapy, systemic conditions, or systemic medications (NSAIDS, steroids, penicillin, vancomycin etc.). Pyuria was also present in samples with contaminated growth, possibly reflecting inadequate collection, storage, or transportation of samples. Thus, pyuria can be detected in non-significant samples, suggesting that leucocytes might be a poor predictor of UTI when using a flowcytometric method.

**Flowcytometry: Evaluation of different cut-off values for bacteria and leucocytes.**
Identifying and selecting the culture negative and contaminated urine samples before culture plating results in reduced number of culture plating and thus lessen workload in the laboratory, as well as improved response time for negative and contaminated cultures. This could reduce over prescription and use of empiric broad-spectrum antibiotics when UTI is unlikely, contributing to reduce overall antibiotic resistance.

We investigated different cut-off values for bacteria and WBCs to find the values that would be most efficient in screening for negative urine samples and minimize false negative results (NPV). Since all samples above a chosen cut-off value would be cultured, specificity and positive predictive value (PPV) is significantly less important than NPV.

As shown in Table 3, the lowest percentage of false negative was found by choosing a cutoff with a low bacterial count. However, this cutoff would yield a very high number of false positive samples, and only reduce culture plating by 8,6%, meaning 91,4% of the samples would need to be culture plated. Increasing cut-off to bacteria $\geq$ 100.000/mL would reduce culture plating by 27,3% but increase false negative samples to 1%.

Adding the parameter of WBCs, culture plating could be further reduced, but false negative samples would increase. As shown in Table 3, using a cutoff with bacteria$\geq$10.000/mL and WBCs $\geq$10/μL, culture plating was reduced by 33%, but false negative would increase to 1,9%. With bacteria $\geq$100.000/mL and WBCs. $\geq$10/μL culture plating could be reduced by 42,1% but false negatives would be 2,7%. Increasing the cut-off for lcc. further, would reduce culture plating but increase false negative samples.

Thus, bacteria alone is the best parameter to minimize false negative findings. Adding WBCs as a parameter does not improve NPV of bacteria alone. The addition of WBCs as a parameter did however increase the number of samples identified as non-significant which would otherwise be cultured. However, this would lead to increasing the number of false negatives. If the aim is to screen efficiently and reduce culture plating (and allowing for some false negative results), bacteria and WBCs. are efficient parameters when used together.

**False negative samples.**    A high NPV is important to minimize false negative samples. For examples, applying a cut-off for positive samples WBCs $\geq$10/μL and bacteria $\geq$100.000/mL, would yield 27 negative samples compared to results from cultivation (Table 4). The bacteria identified in the false negative samples correlates to the bacteria that are normally identified in the laboratory, with *E.coli* being most frequent, followed by *Enterococcus fecalis* and *Klebsiella pneumoniae*.

We also found 23 samples that were positive by flowcytometry but negative when cultured. This is considered due to either non-viable bacteria present in the sample, or bacteria that were slow growing or did not grow in the conditions in which they were incubated. It indicates that urine samples should be incubated in $CO_2$ as recommended in Norwegian national guidelines [12].

**Squamous cells as a parameter of bacterial contamination.**    Detached epithelial cells in the urine may help localize urinary tract disease. According to the European Urinalysis guidelines the appearance of squamous epithelial cells from the outer genitalia or distal urethra serves as a marker of a poor collection technique, except during pregnancy when epithelial cell exfoliation is increased [16]. More than 5 squamous epithelial cells per high power field (HPF) has been used as a cut-off for bacterial contamination [17], whereas other studies do not find

squamous cells to be a good indication for bacterial contamination [18,19] In our study we found that most of the samples with bacterial contamination contained very few squamous cells, and a small number contained very many squamous cells (Fig 3) Thus, we could not find this parameter to be useful to predict bacterial contamination.

## Direct antibiotic susceptibility testing

Direct Antibiotic Susceptibility Testing (dAST) on the positive urine samples could reduce response time for antibiotic profile by up to 24 hours, and thus contribute to reduced empiric use of antibiotic and shorter overall treatment.

A recent study by Rosa et al. concluded that Sysmex UF-5000 showed high diagnostic accuracy in UTI screening with a low rate of false negatives, and that the instrument can predict Gram negative bacteria with a high sensitivity and in high agreement with culture [8]. Direct AST on urine has been evaluated in previous studies [20–24] with promising results. Recently, Wei et al recently suggested the combination of flow cytometry (UF 1000), MALDI-TOF, MS, and VITEK 2 provided a direct, rapid, and reliable identification and AST method for assessing urine samples, especially for Gram negative bacterial infections [25]. As shown in our study, this algorithm could be further improved by using Sysmex UF5000 (that also analyze if the bacteria present are Gram negative and Gram positive) combined with dAST.

**Positive samples with Gram negative bacteria.** As shown in Table 5, of the 81 samples selected for dAST 18 had mixed growth on cultivation. In four samples the mixed growth was only discovered on the cultivation plate, not on the AST plate. This demonstrates the importance of simultaneous culture plating when interpreting AST set up directly on the urine.

In the 56 samples with monobacterial growth the correlation was very good between dAST and Microscan or MALDI-TOF followed by manual AST in the routine (Table 6). We found only 5 discrepancies in a total of 560 tested combinations of antibiotics (Table 7), of which one was VME. For 52 samples the antibiotic profile was similar in dAST and routine and the antibiotic susceptibility profile could be reported the day after arrival of the urine sample.

Analyzing the urine samples on flowcytometry with the bacteria information flag on the evening of arrival also makes it possible to report a preliminary result. A urine sample could be reported as probably positive, with probably Gram positive or probably Gram negative bacteria or both, assisting in the choice of empiric antibiotic for the patient with clinical UTI. In many cases full antibiotic profile could be reported already the next day.

**Positive samples with Gram positive bacteria.** Positive samples with Gram positive bacteria are not suitable for dAST as flowcytometry does not accurately identify urine samples containing one type of gram positive bacteria. The material was, however, very limited with only 25 samples. More than 50% of the samples showed mixed growth/contamination when culture plated, 1came out as yeast a 1 as a Gram negative bacterium. Two samples showed no growth when cultured, this could represent non-viable or slow growing bacteria.

**Identifying samples available for direct AST in daily routine.** Applying a chosen cut-off value for bacteria and WBCs combined with the Gram positive/Gram -negative flag would detect the positive samples containing Gram negative bacteria available for dAST.

In our study, applying a cut-off for positive samples WBCs >10/μL and bacteria >100.000/ mL combined with the Gram positive/Gram negative flag would identify 295 positive samples containing Gram negative bacteria suitable for dAST.

On cultivation of these samples 4 showed no growth, 71 showed mixed growth/contamination and 160 were culture positive. Of these, 155 had significant monomicrobial growth of a Gram negative bacteria and was reported from the laboratory with ASP. In addition, another 60 samples were reported as positive with ASP with a comment that contamination was possible. Thus, of

the 295 samples identified by flowcytometry as positive with Gram negative bacteria, 220 were reported as positive or positive with possible contamination and ASP from the laboratory.

Of the 192 culture positive samples with Gram negative bacteria in our study this algorithm identified 155 (80,1%). Thus 80,1% of culture positive samples containing Gram negative bacteria (and 69,5% of all positive samples) could be reported with ASP the next day, 24 hours earlier than the current routine that uses culture.

We also evaluated if relevant clinical information could be used as a criterium to select samples for dAST for the 295 samples analyzed as positive on Sysmex UF-5000 containing Gram negative bacteria, when compared to the results of cultivation. For 64,8% (46/71) of the contaminated samples there was relevant clinical symptoms of UTI., 64,4% ((103/160) of the positive samples and 55% (33/60) of the samples that were reported as positive but with possible contamination. Thus, we could not find that clinical information could be used as a parameter to consider samples for dAST in our laboratory.

The major drawback with dAST on the urine samples is that it remains a manual procedure with a high percentage of samples that have to be excluded. Of the 295 samples that were suitable for dAST, 71 were clearly contaminated and 5 turned out to be Gram positive bacteria when culture plated, thus 76/295 samples (25%) would be excluded on interpretation.

## Algorithm for use of automated flowcytometry in the diagnosis of UTI

As shown in Fig 4, the urine samples will be analyzed with flowcytometry on the evening of arrival. Negative urine samples will be reported in the evening. All samples that are analyzed as positive on flowcytometry will be culture plated in the routine. The positive samples containing only Gram negative bacteria will be set up as dAST so it can be reported the following day. Special urine samples (e.g. from pregnant woman or bladder puncture) would be culture plated regardless of the result on flowcytometry as they are interpreted differently [12]. Applying a cut-off for positive samples bacteria >100.000/mL and WBCs >10/μL 42,1% of the samples could be reported the same day and 53,8% could be reported the following day. Only 4,1% would be reported on day three, two days after the samples were received.

For the positive samples containing only Gram negative bacteria results of cultivation and dAST will be interpreted together the following day. If monomicrobial growth, the bacteria will be identified by MALDI-TOF, and if identified as a significant pathogen with significant growth [12] ASP is reported. The mixed/contaminated samples will be reported on the day after arrival (as is done today). For the culture positive samples containing Gram positive bacteria considered to be significant AST will be set up and reported after 48 hours, as today. The workload in the laboratory will be reduced depending on the chosen cut-off vales for WBCs and bacteria on flowcytometry, as shown in Table 3.

Recent studies have also evaluated the incubation time for antimicrobial susceptibility testing by disc diffusion for Enterobacteriaceae with results indicating that early reading of inhibition zones to 8–10 hours after incubation is feasible and accurate [22,25,26].Further studies are required, but if incorporated this could reduce the response times from the laboratory even further.

## Conclusion

We found that using flowcytometry offers advantages for UTI diagnosis. Depending on the chosen cut-off values for leucocytes and bacteria, screening for negative urine samples on the day of arrival reduces culture plating and workload, and results in shorter response time for the negative samples. This could reduce use and over prescription of empiric antibiotics, thus contribute to reduce overall antibiotic resistance. The preferred cut-off might vary between laboratories depending on the patient population and an evaluation of what is an acceptable number of false

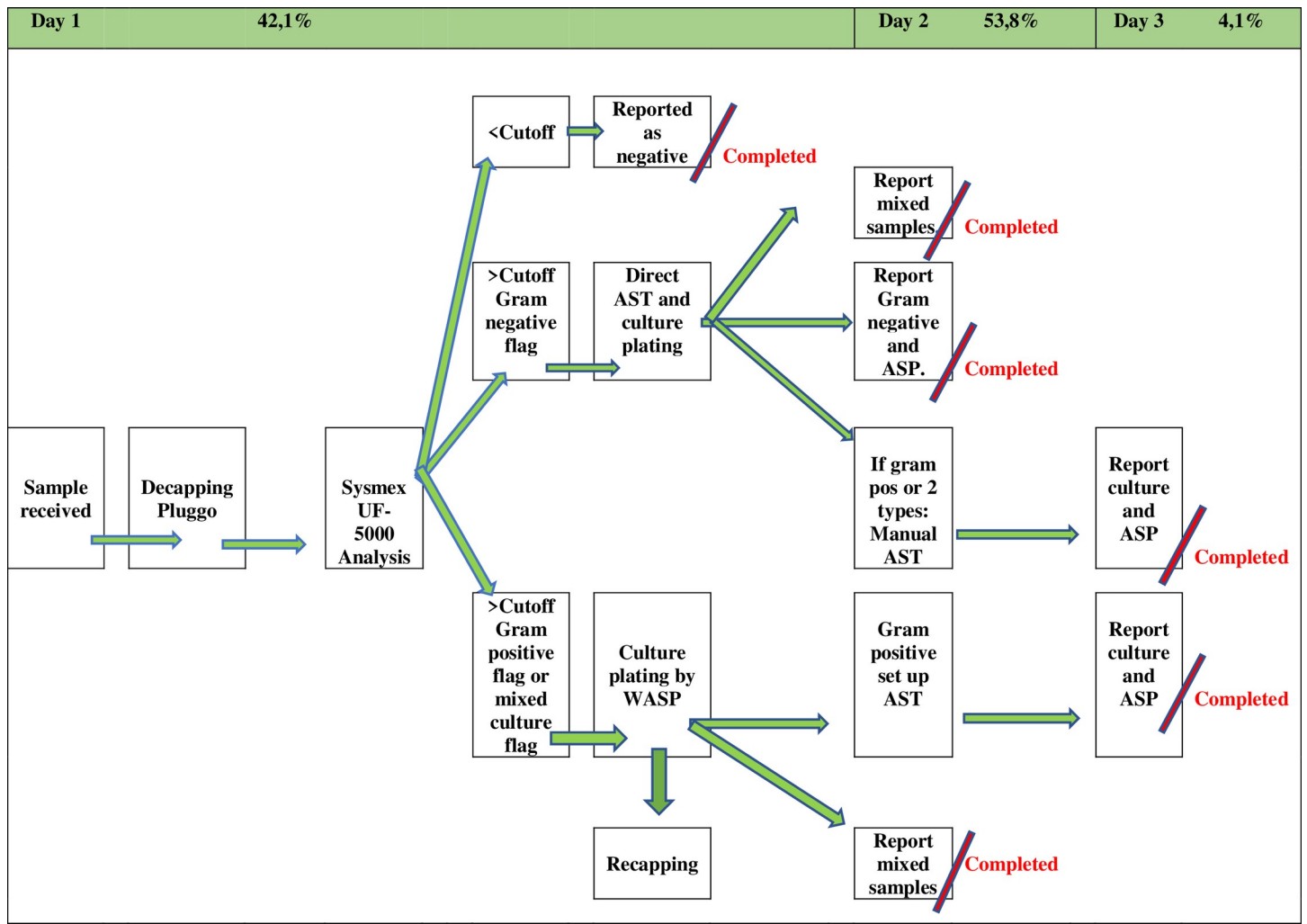

**Fig 4. Algorithm of UTI diagnosis using flowcytometry with reported samples on different days.**

negative samples. If the aim is to screen efficiently and reduce culture plating, bacteria and leucocytes are efficient parameters when used together. If the aim is to reduce culture plating without and minimizing the number of false negative results only bacteria is the preferred parameter.

The bact. flag predicts positive samples containing Gram negative bacteria for dAST. After excluding contaminated/mixed samples and identifying the bacteria the following day, ASP can be reported the day after arrival. For the positive samples containing Gram negative bacteria there was good concordance between dAST and AST in routine with only 5 errors, including one major error in 560 tested combinations. For the remaining 52/56 samples dAST was identical to routine AST. For positive samples containing Gram positive bacteria the results were not convincing.

We did not find any correlation between epithelial cells and contamination.

## Supporting information

**S1 Dataset. Flowcytometric and culture findings.**
(XLSX)

## Acknowledgments

We thank Sysmex Corporation for lending us a Sysmex UF-5000 flowcytometer. We also thank Einar Svartsund for skilled project management, and Amir Moghaddam and Michael Sovershaev for valuable comments and revising the paper. The study could not have been accomplished without support from Fürst Medical Laboratory.

## Author Contributions

**Conceptualization:** Hanne Margrethe Gilboe, Trygve Tjade, Trond Egil Ranheim, Peter Gaustad.

**Data curation:** Trygve Tjade, Johan Bjerner.

**Formal analysis:** Trygve Tjade, Johan Bjerner, Peter Gaustad.

**Investigation:** Hanne Margrethe Gilboe, Olaug Marie Reiakvam, Linda Aasen.

**Methodology:** Hanne Margrethe Gilboe, Olaug Marie Reiakvam, Trygve Tjade, Trond Egil Ranheim, Peter Gaustad.

**Project administration:** Hanne Margrethe Gilboe.

**Supervision:** Hanne Margrethe Gilboe, Olaug Marie Reiakvam, Trygve Tjade, Trond Egil Ranheim, Peter Gaustad.

**Validation:** Hanne Margrethe Gilboe, Olaug Marie Reiakvam, Linda Aasen, Trygve Tjade, Trond Egil Ranheim, Peter Gaustad.

**Visualization:** Hanne Margrethe Gilboe, Olaug Marie Reiakvam, Trygve Tjade, Peter Gaustad.

**Writing – original draft:** Hanne Margrethe Gilboe, Olaug Marie Reiakvam, Linda Aasen.

**Writing – review & editing:** Hanne Margrethe Gilboe, Olaug Marie Reiakvam, Trygve Tjade, Johan Bjerner, Trond Egil Ranheim, Peter Gaustad.

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
