## [Decision Letter · Decision Letter 0]

28 Apr 2021

PONE-D-21-01131

Rapid diagnosis and reduced workload for Urinary Tract Infection using flowcytometry combined with direct Antibiotic Susceptibility Testing

PLOS ONE

Dear Dr. Gilboe,

Thank you for submitting your manuscript to PLOS ONE. After careful consideration, we feel that it has merit but does not fully meet PLOS ONE’s publication criteria as it currently stands. Therefore, we invite you to submit a revised version of the manuscript that addresses the points raised during the review process.

Clarifications on sample acceptability criteria and algorithm regarding mixed cultures needed. 

We look forward to receiving your revised manuscript.

Kind regards,

Iddya Karunasagar

Academic Editor

PLOS ONE

Additional Editor Comments:

The reviewer has raised some points that need clarifications. Justification and reference for accepting samples collected in boric acid at day 4 should be provided.

Journal Requirements:

Reviewers' comments:

Reviewer's Responses to Questions

**Comments to the Author**

1. Is the manuscript technically sound, and do the data support the conclusions?

Reviewer #1: Yes

2. Has the statistical analysis been performed appropriately and rigorously? 

Reviewer #1: Yes

3. Have the authors made all data underlying the findings in their manuscript fully available?

Reviewer #1: Yes

4. Is the manuscript presented in an intelligible fashion and written in standard English?

Reviewer #1: Yes

5. Review Comments to the Author

Reviewer #1: Dear Authors

This is a well written study on the use of Sysmex US 5000 in building an algorithm for rapid processing of urine cultures.

The authors have studied 1000 patients with the routine culture and using sysmex 5000 to screen samples for the presence of significant bacterial counts. The study is well designed and the algorithm can help high throughput labs reduce workload and reduce use of plates for urine culture.

A few minor details that may add value to the paper is

1. Did the authors look at history of antibiotics in those with sterile pyuria ? Could genitourinary tuberculosis be considered in these patients, especially if immunocompromised. I understand the incidence of tuberculosis is very low in Norway, but a clinical suspicion has helped us many a times to identify this problem.

2. What was the acceptability criteria for urine samples collected in boric acid containers? Usually samples beyond 72 hours are not accepted. Why were samples processed even 4 days after collection. Further, did the authors look at the WBC breakdown in samples that were collected later. Did that impact results especially when analysing bacterial counts along with WBC counts?

3. The authors clearly state that Gram positive organisms are yeast do not correlate well with culture in the proposed algorithm. Same goes with mixed cultures. If there is some data on clinical symptoms and the reason for the cultures the algorithm may be further improved. Authors may consider the same.

6. PLOS authors have the option to publish the peer review history of their article (what does this mean?). If published, this will include your full peer review and any attached files.

Reviewer #1: No

---

## [Author Response · Author response to Decision Letter 0]

12 Jun 2021

Sample acceptability and algorithm regarding mixed cultures are included in the manuscipt. 

Sample acceptability: Fürst Medical Laboratory recommends culture plating and examination of a urine sample in boric acid within 48 hours after voiding, in concordance with European Urinalysis Guideline. Clarification on sample acceptability is included in the manuscript under Method section. 

Algorithm regarding mixed cultures: Mixed cultures contain 3 or more strains of bacteria, either Gram positive, Gram negative or both (also included in the manuscript).

Justification for accepting samples in boric acit day 4: Fürst Medical Laboratory recommends culture plating and examination of a urine sample in boric acid within 48 hours after voiding, in concordance with European Urinalysis Guideline. However, as transportation time from rural parts of Norway can be delayed (especially on weekends and holidays) samples are considered for culture plating and examination if it contains information abut clinical infection or/and antibiotics are initiated, as these samples are difficult to reproduce. If the urine sample is culture plated and examined, the clinician will receive a comment that the result is uncertain as the samples was not culture plated within the recommended time frame.

Reference list is rewieved. None of cited articles are retracted. Changes to reference list: Reference 10: Translation to English and details on place of publication has been added. The link has been removed as only the most recent version of the antibiotic panel is table is available from the Norwegian Working Group on Antibiotics’ web site. 

Reference 11: Place of publication has been added. 

Reference 12: List of editors, translation from Norwegian to English, place of publication and page numbers have been added. 

Reference 13: Date of latest update and date of citation have been added. 

Reference 17: English translation has been added. 

None of the cited papers have been retraced. 

Style requirements: manuscript edited and file names edited to comply wih style requirements. 

Data availability: We have uploaded the minimal anonymized data set necessary to replicate the study findings as Supporting Information. We have found that there are no ethical or legal restrictions on sharing a de-identified data set. 

Data not shown: We have removed the phrase as this was not a core part of the research being presented in our study.

Rewiever 1:

1. Did the authors look at history of antibiotics in those with sterile pyuria? No, we did not have this clinical information. 

Could genitourinary tuberculosis be considered in these patients, especially if immunocompromised. I understand the incidence of tuberculosis is very low in Norway, but a clinical suspicion has helped us many a times to identify this problem. Genitourinary tuberculosis is very rare in Norway. If there is a clinical suspicion that a patient has genitourinary tuberculosis, and the clinician informs the laboratory of this, the sample is forwarded to a laboratory that analyzes for this. 

2. What was the acceptability criteria for urine samples collected in boric acid containers? Usually samples beyond 72 hours are not accepted. Why were samples processed even 4 days after collection. Fürst Medical Laboratory recommends culture plating and examination of a urine sample in boric acid within 48 hours after voiding, in concordance with European Urinalysis Guideline. However, as transportation time from rural parts of Norway can be delayed (especially on weekends and holidays) samples are considered for culture plating and examination if it contains information abut clinical infection or/and antibiotics are initiated, as these samples are difficult to reproduce. If the urine sample is culture plated and examined, the clinician will receive a comment that the result is uncertain as the samples was not culture plated within the recommended time frame. 

Further, did the authors look at the WBC breakdown in samples that were collected later. Did that impact results especially when analyzing bacterial counts along with WBC counts? This is a very relevant comment and could have been done. Unfortunately, we did not examine this. However there was only 4 samples that arrived four days after collection so the material was very limited. 

3. The authors clearly state that Gram positive organisms are yeast do not correlate well with culture in the proposed algorithm. Same goes with mixed cultures. If there is some data on clinical symptoms and the reason for the cultures the algorithm may be further improved. Authors may consider the same. Yes, we did examine if clinical information and indication for the urine sample could be a useful indication to do direct antibiotic susceptibility testing on the samples. However, only 55-65 % of received urine samples contain relevant clinical information, so it is difficult to use clinical information systematically in any algorithm.

---

## [Decision Letter · Decision Letter 1]

21 Jun 2021

Rapid diagnosis and reduced workload for Urinary Tract Infection using flowcytometry combined with direct Antibiotic Susceptibility Testing

PONE-D-21-01131R1

Dear Dr. Gilboe,

We’re pleased to inform you that your manuscript has been judged scientifically suitable for publication and will be formally accepted for publication once it meets all outstanding technical requirements.

Kind regards,

Iddya Karunasagar

Academic Editor

PLOS ONE

Additional Editor Comments (optional):

All reviewer comments have been addressed.

Reviewers' comments:

Reviewer's Responses to Questions

**Comments to the Author**

1. If the authors have adequately addressed your comments raised in a previous round of review and you feel that this manuscript is now acceptable for publication, you may indicate that here to bypass the “Comments to the Author” section, enter your conflict of interest statement in the “Confidential to Editor” section, and submit your "Accept" recommendation.

Reviewer #1: All comments have been addressed

2. Is the manuscript technically sound, and do the data support the conclusions?

Reviewer #1: Yes

3. Has the statistical analysis been performed appropriately and rigorously? 

Reviewer #1: Yes

4. Have the authors made all data underlying the findings in their manuscript fully available?

Reviewer #1: Yes

5. Is the manuscript presented in an intelligible fashion and written in standard English?

Reviewer #1: Yes

6. Review Comments to the Author

Reviewer #1: Dear Authors

All concerns have been adequately addressed. For those concerns that could not be addressed in the present manuscript, the numbers are too small to make a difference. Hence the manuscript can be accepted.

7. PLOS authors have the option to publish the peer review history of their article (what does this mean?). If published, this will include your full peer review and any attached files.

Reviewer #1: **Yes: **Anusha Rohit

---

## [Editor Report · Acceptance letter]

25 Jun 2021

PONE-D-21-01131R1 

Rapid diagnosis and reduced workload for Urinary Tract Infection using flowcytometry combined with direct Antibiotic Susceptibility Testing 

Dear Dr. Gilboe:

I'm pleased to inform you that your manuscript has been deemed suitable for publication in PLOS ONE. Congratulations! Your manuscript is now with our production department. 

Kind regards, 

on behalf of

Dr. Iddya Karunasagar 

Academic Editor

PLOS ONE